# Local emergence in Amazonia of *Plasmodium falciparum k13* C580Y mutants associated with *in vitro* artemisinin resistance

Luana C Mathieu[1,2], Horace Cox[3], Angela M Early[4,5†], Sachel Mok[6†], Yassamine Lazrek[1], Jeanne-Celeste Paquet[6], Maria-Paz Ade[7], Naomi W Lucchi[8], Quacy Grant[3], Venkatachalam Udhayakumar[8], Jean SF Alexandre[9], Magalie Demar[10,11], Pascal Ringwald[12], Daniel E Neafsey[4,5‡], David A Fidock[6,13‡], Lise Musset[1*]

[1]Laboratoire de parasitologie, Centre Nationale de Référence du Paludisme, World Health Organization Collaborating Center for surveillance of antimalarial drug resistance, Institut Pasteur de la Guyane, Cayenne, French Guiana; [2]Ecole Doctorale n°587, Diversités, Santé, et Développement en Amazonie, Université de Guyane, Cayenne, French Guiana; [3]Ministry of Public Health, Georgetown, Guyana; [4]Broad Institute of MIT and Harvard, Cambridge, United States; [5]Department of Immunology and Infectious Diseases, Harvard T.H. Chan School of Public Health, Boston, United States; [6]Department of Microbiology and Immunology, Columbia University Irving Medical Center, New York, United States; [7]Department of Communicable Diseases and Environmental Determinants of Health, Pan American Health Organization/World Health Organization, Washington, United States; [8]Malaria Branch, Division of Parasitic Diseases and Malaria, Center for Global Health, Centers for Disease Control and Prevention, Atlanta, United States; [9]Pan American Health Organization, Georgetown, Guyana; [10]Service de Maladies Infectieuses et Tropicales, Centre Hospitalier Andrée Rosemon, Cayenne, French Guiana; [11]Ecosystèmes Amazoniens et Pathologie Tropicale (EPAT), EA3593, Université de Guyane, Cayenne, French Guiana; [12]Global Malaria Program, World Health Organization, Geneva, Switzerland; [13]Division of Infectious Diseases, Department of Medicine, Columbia University Irving Medical Center, New York, United States

**\*For correspondence:**
lisemusset@gmail.com

[†]These authors contributed equally to this work
[‡]These authors also contributed equally to this work

**Abstract** Antimalarial drug resistance has historically arisen through convergent *de novo* mutations in *Plasmodium falciparum* parasite populations in Southeast Asia and South America. For the past decade in Southeast Asia, artemisinins, the core component of first-line antimalarial therapies, have experienced delayed parasite clearance associated with several *pfk13* mutations, primarily C580Y. We report that mutant *pfk13* has emerged independently in Guyana, with genome analysis indicating an evolutionary origin distinct from Southeast Asia. *Pfk13* C580Y parasites were observed in 1.6% (14/854) of samples collected in Guyana in 2016–2017. Introducing *pfk13* C580Y or R539T mutations by gene editing into local parasites conferred high levels of *in vitro* artemisinin resistance. *In vitro* growth competition assays revealed a fitness cost associated with these *pfk13* variants, potentially explaining why these resistance alleles have not increased in frequency more quickly in South America. These data place local malaria control efforts at risk in the Guiana Shield.

**eLife digest** All recommended treatments against malaria include a drug called artemisinin or some of its derivatives. However, there are concerns that *Plasmodium falciparum*, the parasite that causes most cases of malaria, will eventually develop widespread resistance to the drug. A strain of *P. falciparum* partially resistant to artemisinin was seen in Cambodia in 2008, and it has since spread across Southeast Asia. The resistance appears to be frequently linked to a mutation known as *pfk13* C580Y.

Southeast Asia and Amazonia are considered to be hotspots for antimalarial drug resistance, and the *pfk13* C580Y mutation was detected in the South American country of Guyana in 2010. To examine whether the mutation was still circulating in this part of the world, Mathieu et al. collected and analyzed 854 samples across Guyana between 2016 and 2017. Overall, 1.6% of the samples had the *pfk13* C580Y mutation, but this number was as high as 8.8% in one region. Further analyses revealed that the mutation in Guyana had not spread from Southeast Asia, but that it had occurred in Amazonia independently.

To better understand the impact of the *pfk13* C580Y mutation, Mathieu et al. introduced this genetic change into non-resistant parasites from a country neighbouring Guyana. As expected, the mutation made *P. falciparum* highly resistant to artemisinin, but it also slowed the growth rate of the parasite. This disadvantage may explain why the mutation has not spread more rapidly through Guyana in recent years.

Artemisinin and its derivatives are always associated with other antimalarial drugs to slow the development of resistance; there are concerns that reduced susceptibility to artemisinin leads to the parasites becoming resistant to the partner drugs. Further research is needed to evaluate how the *pfk13* C580Y mutation affects the parasite's response to the typical combination of drugs that are given to patients.

## Introduction

Malaria is an important parasitic disease that causes a high level of mortality worldwide. In 2018, malaria was estimated to have caused 405,000 deaths, most of them attributable to the virulent *Plasmodium falciparum* parasite species (*World Health Organization, 2019*). Most malaria cases occur in Sub-Saharan Africa, but Southeast Asia and South America are also affected. Since 2001, Artemisinin-based Combination Therapies (ACTs) have been the recommended first-line therapy for *P. falciparum* infection for almost all malaria-endemic areas (*World Health Organization, 2001*). ACTs combine an artemisinin derivative and one partner drug (most commonly lumefantrine, mefloquine, amodiaquine, piperaquine or pyronaridine). These treatments have contributed to a major reduction in malaria-related mortality and morbidity (*Eastman and Fidock, 2009*; *Carrara et al., 2006*; *Bhatt et al., 2015*). However, in 2008, the first instances of reduced artemisinin efficacy were described in western Cambodia (*Dondorp et al., 2009*; *Ashley et al., 2014*). Subsequent studies have documented the rapid spread of resistance throughout Southeast Asia (*Dondorp et al., 2009*; *Ashley et al., 2014*; *World Health Organization, 2018*). Resistance to artemisinin is partial and affects only rings (*World Health Organization, 2018*). Clinically, this partial resistance trait manifests as a parasite clearance half-life that exceeds 5.5 hr (*WWARN K13 Genotype-Phenotype Study Group, 2019*). This half-life represents the time required to achieve a two-fold reduction of the parasite biomass. Partial resistance also manifests as persistent parasitemia on day three with a complete clearance of parasites following full treatment with an artesunate monotherapy lasting seven days or with an ACT (*World Health Organization, 2018*). An analysis of *P. falciparum* parasites selected for artemisinin resistance *in vitro* as well as parasites from patients experiencing slow parasite clearance led to the identification of a genetic locus linked to resistance: a kelch domain-containing protein located on chromosome 13 (*pfk13*) (*Ariey et al., 2014*). To date, many missense mutations have been described in the BTP/POZ or kelch propeller domain-containing parts of this protein in clinical isolates, however only nine of them have been validated for artemisinin resistance (F446I, N458Y, M476I, Y493H, R539T, I543T, P553L, R561H and C580Y) (*World Health Organization, 2018*;

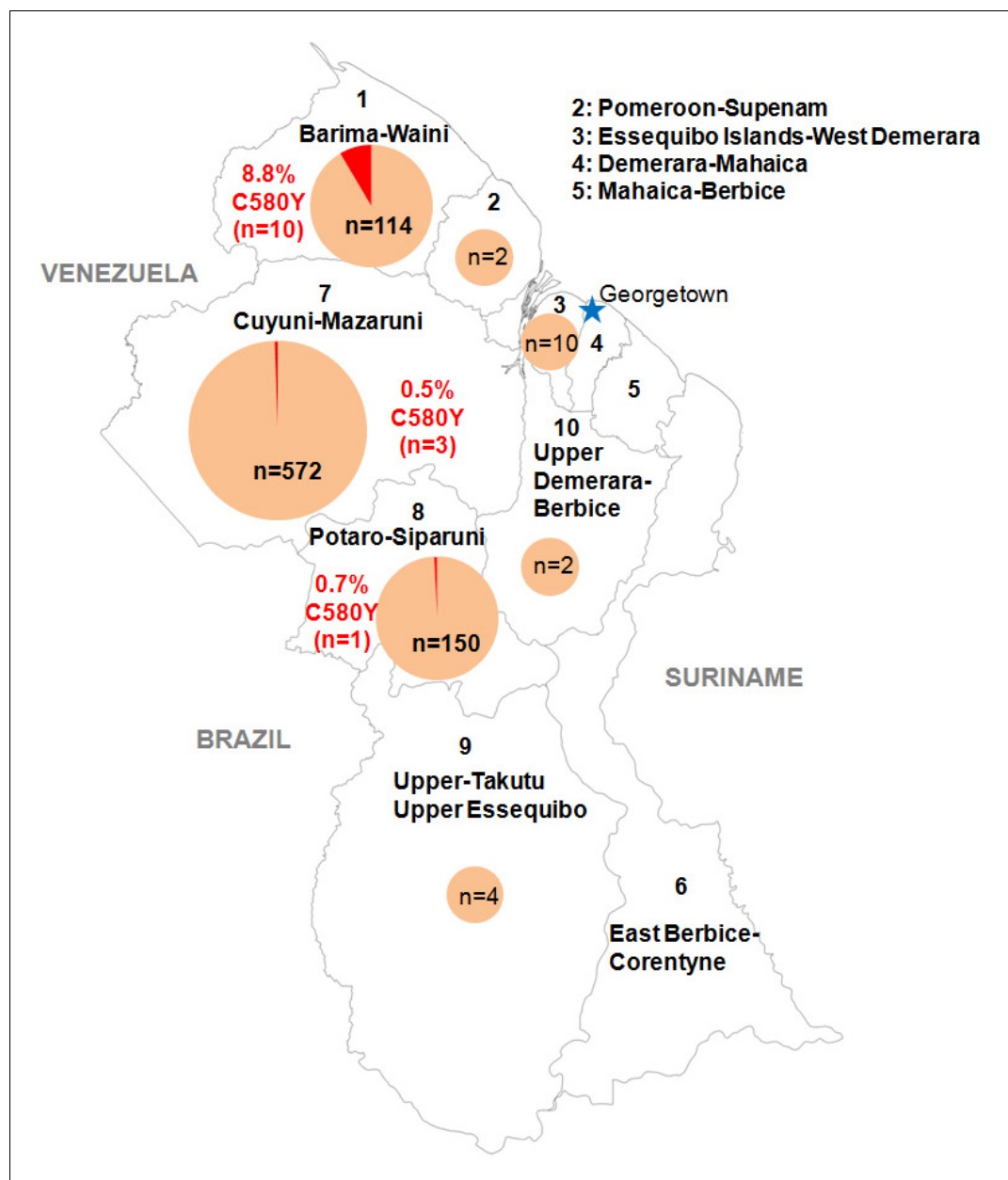

**Figure 1.** Distribution of the *pfk13* C580Y mutant parasites among Guyana regions. Pie charts represent the total number of isolates analyzed per region. Mutants are represented in red.

The online version of this article includes the following source data for figure 1:

**Source data 1.** Number of clinical samples with *pfk13* propeller segment WT or C580Y mutant, analyzed by sampling regions in Guyana from March 2016 to September 2017.

*Ariey et al., 2014*; *Straimer et al., 2015*). A larger number have been associated with delayed parasite clearance (*WWARN K13 Genotype-Phenotype Study Group, 2019*). In most locations, the *pfk13* C580Y mutation has overtaken other resistance-inducing variants, and it is now the most prevalent *pfk13* variant in Southeast Asia (*Ashley et al., 2014*; *Ariey et al., 2014*).

South America has historically been a second hotspot outside Southeast Asia for the evolution of antimalarial drug resistance. For example, chloroquine and sulfadoxine-pyrimethamine resistance evolved simultaneously in both regions (*Wootton et al., 2002*; *Roper et al., 2004*). The Guiana Shield region of South America, which includes Guyana, Suriname, French Guiana and parts of Brazil, Venezuela and Colombia, is important as a potential source of emerging antimalarial drug

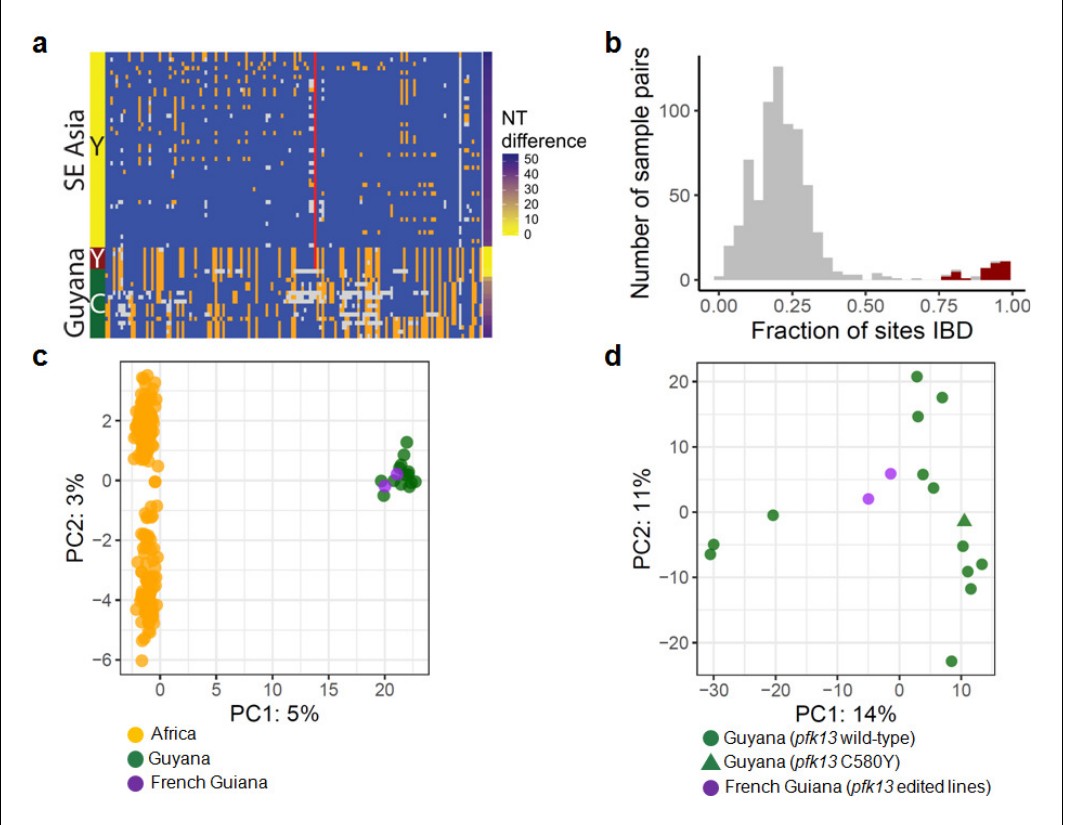

**Figure 2.** Whole-genome sequence analysis of *pfk13* C580Y mutant parasites in Guyana. (a) Comparison of the haplotypic background of *pfk13* C580Y mutant parasites from Guyana, 2016, and Southeast Asia, 2010–2012. Across Pf3k samples from Cambodia, Thailand, and Vietnam, 45 unique C580Y-coding haplotypic backgrounds were identified and compared to haplotypes from Guyana. Columns represent 149 sites containing non-singleton single nucleotide polymorphisms (SNPs) found within a 150 kb segment surrounding the *pfk13* C580Y-coding allele. At a given site, the more common allele is marked blue, the less common allele is orange, and missing calls are grey. The Y-coding variant for codon 580 of *pfk13* is represented by the red blocks; wild-type is blue. Only the five *pfk13* C580Y mutant samples with fewer than 15% missing calls are depicted here. (b) Analysis of relatedness at the whole-genome level among Guyana clones. Pairwise identity-by-descent (IBD) was estimated for all pairs of Guyana samples with high quality whole-genome sequence data (<70% missing calls). Pairwise comparisons between samples exhibiting the *pfk13* C580Y allele are indicated in red, and show uniformly high levels of relatedness, suggesting a single clonal lineage harboring the resistance mutation. (c, d) Principal components analysis of parasites from Guyana or other geographic regions using SNP calls from whole-genome sequence data. (c) The parasites from Guyana and French Guiana form a single cluster when compared with parasites from Africa. (d) The two edited parasite lines from French Guiana are highly similar to the sequenced parasite samples from Guyana including a *pfk13* mutant.

The online version of this article includes the following figure supplement(s) for figure 2:

**Figure supplement 1.** Number of single nucleotide differences between pairs of parasites within different geographic locations.

resistance. In this region, the subsoil is rich in gold and other minerals, leading to extensive mining activities in forested areas. These mining activities favor malaria transmission, particularly *P. falciparum* (*Douine et al., 2016*; *Heemskerk, 2011*). Mining also generates human population movement and inappropriate use of ACTs, as people generally work deep in the forest, far from medical care (*Pribluda et al., 2014*; *Douine et al., 2018*). This erratic use of ACTs, as well as self-medication using ACTs of substandard quality or artemisinin monotherapies, may promote the emergence of resistance to artemisinin.

In Guyana in 2010, 5.1% (5 out of 98) of *P. falciparum*-infected clinical samples, sharing a common haplotype different from their Cambodian counterparts, exhibited the *pfk13* C580Y mutation

(*Chenet et al., 2016*). However, no clinical or *in vitro* resistance phenotypes were measured or associated with the genotype data. The most recent therapeutic efficacy study conducted in Guyana in 2014 with 50 patients failed to identify *pfk13* mutations or delayed parasite clearance time after artemether/lumefantrine treatment (*Rahman et al., 2016*). The sample size, however, was too small to ensure detection of low-frequency resistance mutations. The objective of the present study was to evaluate whether *pfk13* variants (C580Y or others) have persisted in Guyana since 2010, and if so, to evaluate their prevalence in 2016–2017. In the absence of available clinical phenotypes associated with the *pfk13* C580Y variant from patients infected and treated in this region, we evaluated the impact of two *pfk13* mutations, C580Y and R539T, on *in vitro* resistance and the rate of parasite growth in a South American parasite genetic background.

## Results

### Recent circulation of the *pfk13* C580Y mutation on a single clonal background

We genotyped the propeller domain of the *pfk13* gene by Sanger sequencing 854 samples collected between March 2016 and September 2017 from different malaria-endemic regions of Guyana. Fourteen samples bearing the *pfk13* C580Y variant were identified, representing a prevalence of 1.6% ($CI_{95}$0.8–2.5%). The prevalence of mutants was 8.8% ($CI_{95}$3.6–14.0%) in Region 1, 0.7% ($CI_{95}$0.0–2.0%) in Region 8 and 0.5% ($CI_{95}$0.0–1.1%) in Region 7 (*Figure 1*, *Figure 1—source data 1*). We also genotyped the sequence outside of the propeller domain for 283 samples. Among these, 57.2% ($CI_{95}$51.5–63.0%) were the 3D7 reference genotype, 42.0% ($CI_{95}$36.3–47.8%) carried the *pfk13* K189T mutation and 0.7% ($CI_{95}$0.0–1.7%) had a mixed genotype (wild-type/K189T). All samples bearing the *pfk13* C580Y variant also exhibited the *pfk13* K189T mutation, which to date has not been associated with artemisinin resistance.

To understand the origin of the C580Y mutation and examine the genetic relatedness of the mutant strains, we performed whole-genome sequencing (WGS) on thirteen of the fourteen Guyana samples exhibiting the *pfk13* C580Y mutation, as well as 40 comparator samples exhibiting wild-type *pfk13* and collected at comparable locations and times. The variant profile observed in the sequenced samples identified that the *pfk13* C580Y variant arose on a single Guyanese parasite genetic background, and was not imported from Southeast Asia (*Figure 2a*). Deeper examination of the genomic similarity was performed by estimating the proportion of the genome that was identical-by-descent between sample pairs. This analysis revealed an extremely high level of relatedness between sample pairs bearing *pfk13* C580Y (identity by descent (IBD) >0.77) relative to pairwise comparisons containing at least one wild-type *pfk13* sample (*Figure 2b*). This indicates that the parasite lineage on which C580Y arose in Guyana in 2010 probably engaged in limited sexual outcrossing with other parasite lineages in Guyana, despite having persisted for a sufficient duration of time to be observed in multiple regions of the country and to have risen to a non-negligible frequency in the Region 1 population.

In the 75 kb segments flanking the 2016–2017 C580Y-coding variant, WGS identified only two low-quality single nucleotide variants among the 13 mutant samples (*Figure 2a*). We also analyzed *pfk13* C580Y parasites using eight microsatellite loci flanking the *pfk13* gene. Despite missing data for some loci in some samples, two different haplotypes were identified, differing at one locus positioned at −6.36 kb (*Table 1*). The previously identified Guyana A mutant haplotype (*Chenet et al., 2016*) matched the most common haplotype in the 2016–2017 samples, which was markedly distinct from the one observed in *pfk13* Cambodian mutants (*Chenet et al., 2016*). The second 2016 haplotype exhibiting the 280 allele at the locus −6.36 kb has not been previously identified. As replicate genotyping efforts reproduced the allelic variant at locus −6.36, it is likely that a *de novo* microsatellite mutation occurred at locus −6.36 following the origin of the C580Y mutation. The high similarity of the microsatellite haplotypes bearing C580Y between 2010 (*Chenet et al., 2016*) and 2016 suggests that the chromosome 13 resistance haplotype, and perhaps the full clonal lineage, has persisted over this timespan, as several of the allelic markers associated with the C580Y mutation are rare in both studies (*e.g.* allele 277 at marker −6.36: 11%; allele 206 at marker −0.15: 9%; allele 244 at marker 72.3: 6% *Chenet et al., 2016*). We were not able to procure remaining DNA from the samples collected in 2010 to determine whether the entire genomic background has been preserved

**Table 1.** *pfk13* microsatellite analysis of Guyanese and Cambodian isolates.

| Name | Region | Year of collection | −31.9 | −6.36 | −3.74 | −0.15 | K13° | 3.4 | 8.6 | 15.1 | 72.3 |
|---|---|---|---|---|---|---|---|---|---|---|---|
| T145 | 1 | 2016 | 203 | 277 | 170 | 206 | C580Y | 138 | 262 | 144 | 244 |
| T237 | 7 | 2016 | 203 | 277 | 170 | 206 | C580Y | 138 | 262 | 144 | 244 |
| T244 | 1 | 2016 | 203 | 280 | 170 | NA | C580Y | 138 | NA | 144 | NA |
| T305 | 1 | 2016 | 203 | 277 | 170 | 206 | C580Y | 138 | 262 | 144 | 244 |
| T345 | 1 | 2016 | 203 | 277 | 170 | 206 | C580Y | 138 | 262 | 144 | 244 |
| T364 | 1 | 2016 | 203 | 277 | 170 | 206 | C580Y | 138 | 262 | 144 | 244 |
| T378 | 1 | 2016 | 203 | 277 | 170 | 206 | C580Y | NA | 262 | 144 | 244 |
| T385 | 7 | 2016 | 203 | 277 | 170 | 206 | C580Y | 138 | 262 | 144 | 244 |
| T445 | 1 | 2016 | 203 | 277 | 170 | 206 | C580Y | NA | 262 | 144 | 244 |
| T490 | 7 | 2016 | 203 | 280 | 170 | 206 | C580Y | NA | 262 | 144 | 244 |
| T508 | 8 | 2016 | 203 | 277 | 170 | 206 | C580Y | 138 | 262 | 144 | 244 |
| T314 | 1 | 2016 | NA | NA | NA | NA | C580Y | NA | NA | NA | NA |
| T649 | 1 | 2016 | NA | NA | NA | NA | C580Y | NA | NA | NA | NA |
| GUY0183 | 1 | 2017 | ND | ND | ND | ND | C580Y | ND | ND | ND | ND |
| T208 | 7 | 2016 | 203 | 280 | 152 | 192 | WT | 138 | 284 | 138 | 244 |
| T265 | 7 | 2016 | 203 | 280 | 156 | 192 | WT | 102 | 262 | 144 | 238 |
| T317 | 1 | 2016 | NA | NA | 156 | 190 | WT | NA | 262 | 144 | 244 |
| T332 | 7 | 2016 | NA | NA | 154 | 190 | WT | 102 | 262 | 144 | 244 |
| T504 | 8 | 2016 | 203 | 280 | 152 | 192 | WT | 138 | 284 | 138 | 244 |
| T524 | 7 | 2016 | NA | NA | 170 | 192 | WT | 102 | 270 | NA | NA |
| T634 | 8 | 2016 | 205 | 280 | 156 | 192 | WT | 102 | 262 | 144 | 238 |
| T724 | 8 | 2016 | 203 | 280 | 154 | 192 | WT | NA | 284 | 138 | 240 |
| | | | | | | | | | | | |
| Guyana A* | 1 and 7 | 2010 | 203 | 277 | 170 | 206 | C580Y | 138 | 262 | 144 | 244 |
| Guyana B* | 7 | 2010 | 203 | 277 | 170 | 206 | C580Y | 138 | 262 | 144 | 240 |
| | | | | | | | | | | | |
| MRA 1236* | | 2010 | 201 | 283 | 146 | 194 | C580Y | 130 | 286 | 138 | NA |
| MRA 1240* | | 2011 | 201 | 283 | 146 | 194 | R539T | 122 | 264 | 138 | 244 |
| MRA 1241* | | 2011 | 201 | 283 | 146 | 194 | I453T | 130 | 290 | 138 | 244 |
| | | | | | | | | | | | |
| 3D7 | | | 207 | 283 | 164 | 226 | WT | 160 | 274 | 147 | 242 |
| 7G8 | | | 225 | 280 | 158 | 196 | WT | 102 | 262 | 144 | 240 |

°Codons 438–704, NA: No Amplification, ND: Not Done, *: Mutants from **Chenet et al., 2016** according to the new size-assignment for microsatellites.

intact as a clonal lineage between 2010 and 2016. However, the genotypes of the molecular markers for resistance (*pfcrt*, *pfdhps*, *pfdhfr* and *pfmdr1*) were also similar between 2010 and 2016 mutant samples (**Supplementary file 1**).

## Evidence for a Guyana-specific genetic background of *pfk13* C580Y mutants

The observation of the C580Y mutation persisting in only one clonal lineage suggests that genomic background may be an important determinant of the emergence and persistence of *pfk13* propeller mutations in Guyana, as was previously observed in Southeast Asia (**Miotto et al., 2013**; **Miotto et al., 2015**; **Cerqueira et al., 2017**; **Amato et al., 2018**). We therefore compared the genetic background of 53 isolates from Guyana (40 *pfk13* wild-type and 13 *pfk13* C580Y) and the artemisinin-resistant background observed in Southeast Asian parasites, in order to explore whether

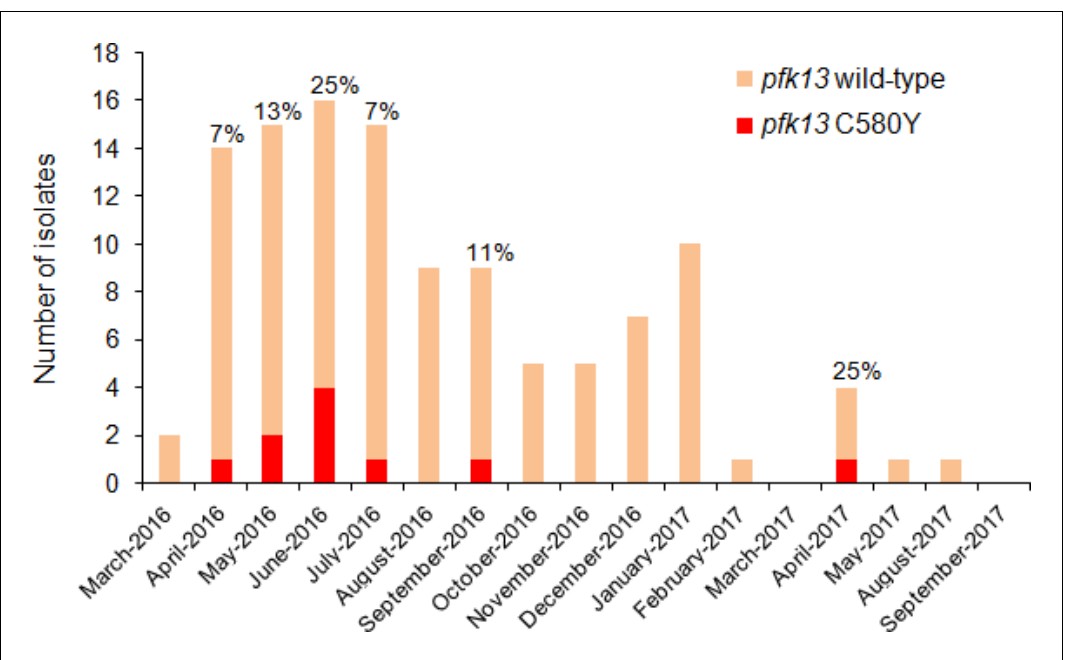

**Figure 3.** Temporal distribution of *pfk13* C580Y mutants in Region 1 of Guyana per month of collection, 2016–2017. The percentage of *pfk13* C580Y mutants for each month of identification is represented above each bar. The online version of this article includes the following source data for figure 3:

**Source data 1.** Number of clinical samples with *pfk13* WT or C580Y mutant alleles in Region 1 of Guyana from March 2016 to September 2017.

partner mutations associated with *pfk13*-mutant parasites may be responsible for the preservation of this clonal lineage (*Miotto et al., 2015*). None of the previously described mutations were observed, including the *pffd* D193Y, *pfcrt* N326S, *pfcrt* I356T, *pfarps* V127M and *pfmdr2* T484I variants. The *pfcrt* gene was of South American origin with a 7G8 haplotype comprising the mutations C72S, K76T, A220S, N326D and I356L, with no particular difference between wild-type and *pfk13* C580Y parasites (*Supplementary file 1*). However, within this candidate gene set, we observed two fixed differences between wild-type and *pfk13* C580Y parasites, both within the gene PF3D7_1252100 (*RON3*) (*Supplementary file 2*). Although overall genotyping rates were low in this gene (successful for 15 out of 53), all genotyped *pfk13* C580Y mutants (n = 5) contained a V1661L-coding variant and all genotyped *pfk13* wild-type parasites (n = 9) carried a synonymous non-reference variant at codon 1801. We also analyzed other molecular markers for resistance (*pfcrt*, *pfdhps*, *pfdhfr* and *pfmdr1*) (*Supplementary file 1*). All *pfk13* C580Y variants were *pfcrt* SVMNT (72-76)/A220S/N326D/C350/I356L (i.e. the *pfcrt* 7G8 haplotype), *pfdhfr* double-mutant (N51I/S108N), *pfdhps* triple-mutant (A437G/K540E/A581G) and *pfmdr1* triple-mutant (Y184F/N1042D/D1246Y). In summary, Guyanese parasites bearing *pfk13* C580Y also exhibited variants in genes previously associated with separate antimalarial drug resistance phenotypes in Southeast Asia.

## A single *pfk13* C580Y clone has fluctuated through time without massive spreading

The highest prevalence of the Guyanese *pfk13* C580Y variant was found in Region 1. This variant was first observed in April 2016 and achieved a maximum prevalence of 25.0% (CI$_{95}$3.8–46.2%) in June 2016 (*Figure 3*, *Figure 3—source data 1*). Thereafter, it was not identified between September 2016 and March 2017 despite a constant transmission level of malaria in this region during this period. To understand these fluctuations in prevalence, we studied the parasite population dynamics in the country. We compared parasites from Guyana to samples from Africa and Southeast Asia (*Supplementary file 3*). At synonymous sites within 4888 genes, pairwise nucleotide diversity ($\pi_{syn}$) in Guyana was $3.4 \times 10^{-4}$, less than half the level observed in Southeast Asia ($7.4 \times 10^{-4}$) and nearly a third of that measured in Africa ($1.0 \times 10^{-3}$). In Guyana, 46 samples had sufficient sequencing

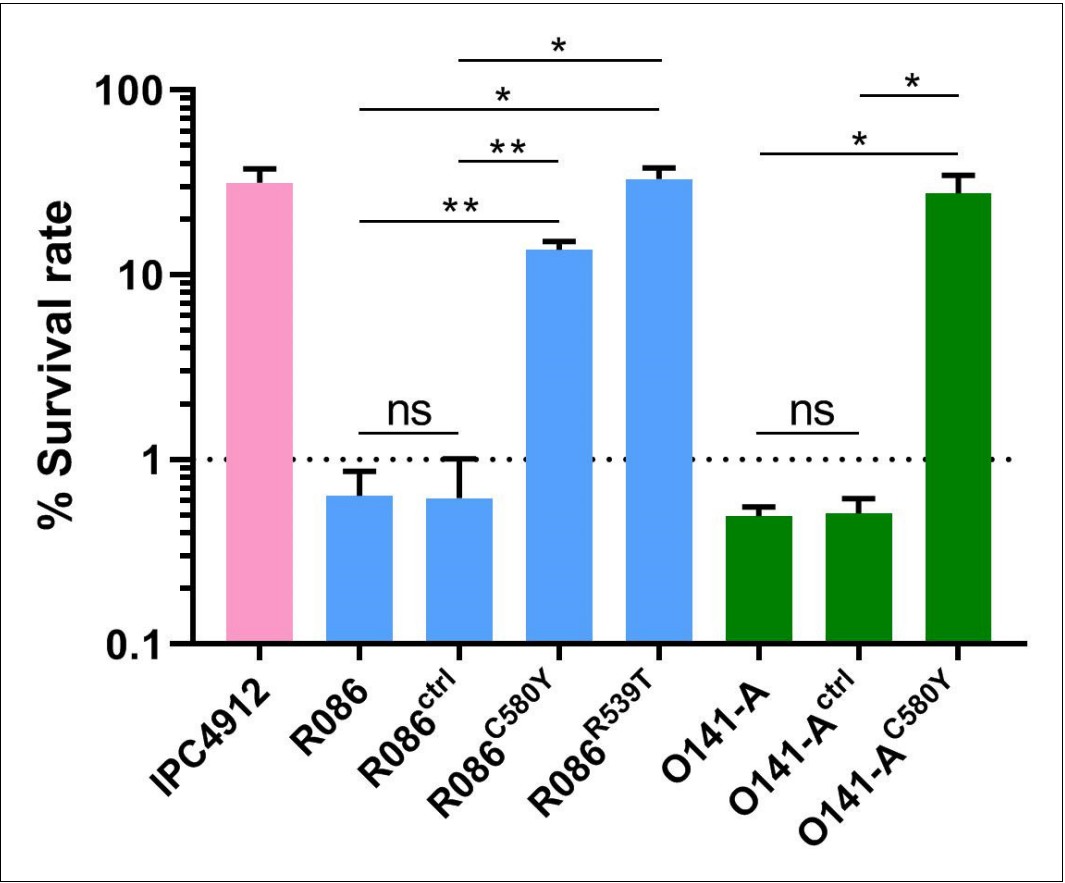

**Figure 4.** Ring-stage Survival Assays in parasites from French Guiana. Data show survival rates of ring-stage parasites (0–3 hr post invasion of human erythrocytes) after a 6 hr pulse of 700 nM DHA, as measured by microscopy 66 hr later. Data illustrate mean ± SEM percent survival from three independent repeats compared with dimethyl sulfoxide (DMSO)-treated parasites as a control for two isolates from French Guiana (O141-A, R086). Parents harbored wild-type *pfk13* allele, and for zinc-finger nuclease edited isogenic parasites, control (ctrl) isolates harbored wild-type *pfk13* allele with silent mutations or *pfk13* mutations (C580Y or R539T). IPC4912, a Cambodian reference strain harboring the I543T *pfk13* mutation was used as a control. A parasite line is considered resistant when the survival rate is greater than 1%. Student's t-test was used to assess significant differences between survival rates of parental and *pfk13*-edited parasites. *p<0.05; **p<0.01; ns: not significant. The online version of this article includes the following source data for figure 4:

**Source data 1.** Survival rates obtained on *pfk13* gene-edited (wild-type, C580Y or R539T) isogenic field isolates from French Guiana.

coverage to assess their complexity of infection (COI), and only one (2.2%) showed evidence of multiple parasite lineages (COI > 1). In contrast, multiclonal infection estimates from Africa and Southeast Asia are routinely higher (*Chang et al., 2017*; *Galinsky et al., 2015*; *Assefa et al., 2014*; *Zhu et al., 2019*). Despite the low nucleotide diversity and small proportion of multiclonal infections, Guyana harbors a relatively high number of distinct parasite lineages and few parasite pairs with identical genomes. We found that only 0.54% of wild-type Guyanese parasite pairs showed evidence of belonging to the same clonal lineage (IBD >0.75).

## *pfk13* C580Y and R539T mutations generate *in vitro* artemisinin resistance in cultured parasites from the Amazonia

To evaluate the impact of the *pfk13* C580Y mutation on artemisinin susceptibility in parasites, we culture-adapted two *P. falciparum* isolates (O141-A and R086) from French Guiana (a neighboring country in the Guiana Shield) and genetically edited these lines using a previously described zinc-finger nuclease (ZFN) based approach (*Straimer et al., 2015*). We also evaluated the impact of the

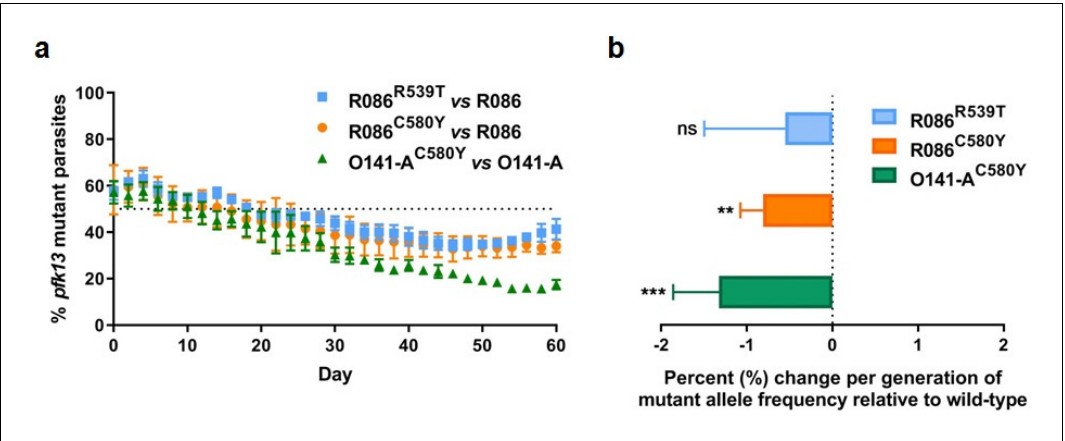

**Figure 5.** Competition growth assays of *pfk13* mutant and wild-type parasites. (a) Frequency of wild-type and mutant parasites in co-culture, as measured by TaqMan allelic discrimination qPCR. Data show the percentage of *pfk13* mutant parasites in the culture over 60 days with sampling every two days. Error bars represent the SEM of *pfk13* mutant allele frequency between the two biological replicates (including two technical replicates for qPCR). A percentage below 50% indicates the mutant was less fit than the isogenic *pfk13* wild-type line. (b) Percentage change per generation of *pfk13* mutant allele frequency relative to wild-type. Data show that *pfk13* mutations confer an *in vitro* fitness cost in both parasite lines. Differences in growth rates were calculated as the percent change in *pfk13* mutant allele frequency averaged over 30 generations. Error bars represent the SEM of percentage growth change between the two biological sampling experiments calculated for every generation in each co-culture. Significance was calculated using the Wilcoxon signed-rank test in every generation across the two biological replicate experiments. **p<0.01, ***p<0.001; ns: not significant.

The online version of this article includes the following source data and figure supplement(s) for figure 5:

**Source data 1.** Proportion of *pfk13* mutants compared to *pfk13* wild-type during 60 days of *in vitro* co-culture.
**Figure supplement 1.** Representative standard curves for qPCR reactions targeting *pfk13* C580/C580Y or *pfk13* R539/R539T allele.
**Figure supplement 1—source data 1.** Data for establishment of standard curves and pre-defined set of mixtures of each *pfk13* allele (C580Y or R539T) used in the optimization of the Taqman allelic discrimination qPCR assays.
**Figure supplement 2.** Reproducibility of TaqMan allelic discrimination qPCR performed on R086^R539T and R086 parasites.
**Figure supplement 2—source data 1.** Data for the reproducibility of the Taqman allelic discrimination assay across qPCR repeats and sampling replicates performed with R086^WT vs R086^R539T parasites.

variant arginine to threonine mutation at codon 539 (R539T), associated with one of the highest *in vitro* artemisinin resistance levels in Asian parasites (*Ariey et al., 2014*; *Straimer et al., 2015*). These two culture-adapted lines both exhibit the *pfcrt* SVMNT haplotype that is associated with chloroquine resistance, but only R086 carries the *pfcrt* C350R mutation that was earlier found to restore chloroquine susceptibility to South American parasites despite the presence of the SVMNT haplotype (*Pelleau et al., 2015*). These parasite lines from French Guiana exhibit a high level of genomic similarity to our genome-sequenced Guyanese parasites (*Figures 2c and d*; *Figure 2—figure supplement 1*). These findings suggest that phenotypes observed in these lines from French Guiana are relevant for understanding the impact of the *pfk13* C580Y variant in Guyanese parasites, none of which have yet been adapted to *in vitro* culture.

We produced isogenic lines expressing the wild-type *pfk13* allele (R086^ctrl, O141-A^ctrl) or the *pfk13* C580Y mutation (R086^C580Y, O141-A^C580Y), or the *pfk13* R539T allele in the case of R086 (R086^R539T). We measured the phenotypic impact of those mutations on *in vitro* artemisinin resistance in edited and parental lines using the Ring-stage Survival Assay (*Witkowski et al., 2013*) that begins with 0–3 hr post-invasion rings (RSA$_{0-3h}$). Introducing the *pfk13* C580Y mutation caused a significant increase in the survival rate (twenty-three fold in R086: 0.6% ± 0.2 for R086 to 13.7% ± 1.5 for R086^C580Y, p=0.006; fifty-five fold in O141-A: 0.5% ± 0.1 for O141-A compared to 27.6% ± 7.0 for O141-A^C580Y, p=0.030, *Figure 4*, *Figure 4—source data 1*). R086^R539T showed a survival rate fifty-five times higher than the parental line (33.0% ± 5.0 compared to 0.6% ± 0.2, p=0.011). Control

assays found no significant difference in the survival rate of the parental *vs.* isogenic control lines (0.6% ± 0.2 for R086, 0.6% ± 0.4 for R086$^{ctrl}$*p*=0.968, 0.5% ± 0.1 for O141-A and O141-A$^{ctrl}$*p* = 0.899) (*Figure 4*, *Figure 4—source data 1*).

## *Pfk13* C580Y and R539T have a fitness impact on parasites depending on the genetic background

To assess the *in vitro* fitness of the *pfk13* mutants relative to the wild-type isogenic lines, we performed a competitive growth assay by co-culturing each pair of mutant and wild-type isogenic parasite lines and measuring the *pfk13* allele frequencies over 60 days (~30 asexual generations). This was achieved using a highly sensitive and robust Taqman allelic discrimination real-time PCR (qPCR) assay (*Figure 5—figure supplement 1*, *Figure 5—figure supplement 2*, *Figure 5—figure supplement 1—source data 1*, *Figure 5—figure supplement 2—source data 1*) that was able to accurately quantify the different proportions of the mutant alleles in these samples. Results showed a modest growth deficit in *pfk13* mutants compared to the wild-type parent for both the R086 and O141-A lines, suggesting that the *pfk13* C580Y and R539T mutations negatively impact parasite growth. Over the 60 day period, we observed an 18% reduction in the frequency of the R539T mutant for the R086 line, and larger significant reductions of 24% and 40% in allele frequencies of the C580Y mutants of R086 and O141-A compared to their respective isogenic wild-type counterparts (*Figure 5a*, *Figure 5—source data 1*). This reflected an average reduced growth rate of −0.6%, −0.8% and −1.3% per 48 hr generation of the R086$^{R539T}$, R086$^{C580Y}$ and O141-A$^{C580Y}$ mutant lines, respectively, across 30 generations (*Figure 5b*, *Figure 5—source data 1*). In addition to observing that the C580Y mutation conferred a higher fitness cost compared to the R539T mutation, we also saw that the parasite genetic background contributed to the severity of the fitness deficit, as the growth defect in the C580Y mutation was substantially more pronounced in O141-A compared with R086 parasites (*Figure 5*, *Figure 5—source data 1*).

## Discussion

Results presented herein confirm the *de novo* emergence and long-term persistence of the *pfk13* C580Y artemisinin resistance mutation in Guyana, South America. Six years after their first identification, *pfk13* C580Y mutant parasites continue to circulate in Guyana at low prevalence (1.6%; 14/854). Parasites bearing this mutation belong to a single clonal lineage and are of autochthonous origin. Gene editing studies on French Guianan parasites, closely related to parasites from Guyana, showed that the *pfk13* C580Y mutation is able to mediate artemisinin resistance *in vitro* at levels observed in Southeast Asian parasites harboring this mutation. An adverse impact of this *pfk13* mutation on asexual blood stage growth rates of *pfk13*-edited French Guianan parasite lines in culture was also observed, providing evidence of a fitness cost similar to that previously described for Southeast Asian lines (*Nair et al., 2018*; *Li et al., 2019*).

During the 1990s, analyses of resistant parasites retrospectively suggested the simultaneous emergence of resistance to chloroquine and sulfadoxine/pyrimethamine in parasites from Southeast Asia and the Amazonian region of South America (*Wootton et al., 2002*; *Roper et al., 2004*; *Vinayak et al., 2010*). With artemisinin, the same scenario is occurring with evidence of an independent emergence of resistance mutation in South America. For artemisinin resistance, there is an opportunity to dissect the first steps of resistance selection in South America in 'real time', relying on genetic markers and phenotypic assays (genotypic surveys, therapeutic efficacy studies, and *in vitro* phenotyping through the Ring-stage Survival Assay (RSA$_{0-3h}$)) developed in Southeast Asia (*Ariey et al., 2014*; *Witkowski et al., 2013*). These genetic and phenotypic tools can help address the critical questions of why *in vitro* resistance to artemisinin has emerged in Region 1 in Guyana, a country with fewer than 20,000 malaria cases in 2018, and why this resistance is not increasing more quickly. In 2004, Guyana was one of the first South American malaria-endemic countries to adopt and implement artemether-lumefantrine (Coartem), the current first-line therapy against *P. falciparum*. Six years later, in 2010, *pfk13* C580Y mutants were first identified. Until now, no further signals of artemisinin resistance or *pfk13* mutations had been identified, despite studies incorporating *pfk13* genotyping in Guyana (n = 50), Suriname (n = 40) and French Guiana (n = 198) (*Rahman et al., 2016*; *Ménard et al., 2016*; *Chenet et al., 2017*). Other than *pfk13* C580Y, no other *pfk13* propeller domain mutations have been observed in Guyana. In Southeast Asia, two molecular epidemiological

profiles are currently observed. In Western Cambodia, one C580Y linage (KEL1) has displaced other mutations and has rapidly become the dominant *pfk13* genotype in that population (>80%) (*Amato et al., 2018*; *Takala-Harrison et al., 2015*). This lineage has merged with a multicopy *plasmepsin 2* and *3* lineage (PLA1) that is associated with piperaquine resistance (*Amato et al., 2018*). Nowadays, this co-lineage has colonized northeastern Thailand and southern Laos (*Imwong et al., 2017*). On the other side, in western Thailand, at the border with Myanmar, a six-year period was required for distinct mutant *pfk13* parasite lineages to collectively reach a population prevalence of 20% (*Amato et al., 2018*; *Anderson et al., 2017*). At the present time, a patchwork of *pfk13* genotypes co-circulates in that population and the prevalence of C580Y lineages fluctuates around 20–30% (*Kobasa et al., 2018*). In Myanmar, *pfk13* mutants are more diverse and the F446I mutation presently dominates the *pfk13* mutant parasite population in certain sites, with this mutation being associated with an intermediate resistance phenotype (*Han et al., 2020*; *Bonnington et al., 2017*).

Given this history of *pfk13* propeller domain mutations in Southeast Asia, beginning with a soft sweep and transitioning to a hard sweep, we might have expected an increase in the prevalence of *pfk13* C580Y mutant parasites during this 2010-2016/2017 period in Guyana. However, the situation in this part of the world differs from Southeast Asia. First, efforts to monitor *pfk13* mutations in Guyana were not systematically conducted between 2010 and 2017. Sample size could therefore explain why the C580Y mutation was only sporadically observed. Nonetheless, the present study indicates that the mutation has not drastically increased in the parasite population. Differences in the resistance profile to artemisinin partner drugs could also account for the heterogeneity in *pfk13* mutational trajectories in Southeast Asia vs. Guyana, a region where markers of resistance have historically reached fixation (*Pelleau et al., 2015*; *Legrand et al., 2012*). In Southeast Asia, the rapid spread of artemisinin-resistant genomic lineages can be explained in part by a multidrug-resistant profile, which includes *pfk13* variants as well as mutations conferring resistance to the partner drug piperaquine (*Amato et al., 2018*; *Ross et al., 2018*; *Kim et al., 2019*). Our genomic analysis of a panel of Guyanese isolates identified the common South American profile for known drug resistance markers (*pfcrt*, *pfdhps*, *pfdhfr* and *pfmdr1*) in both mutant and wild-type *pfk13* samples (*Wootton et al., 2002*; *Roper et al., 2004*; *Vinayak et al., 2010*; *Legrand et al., 2012*). In the Guiana Shield, lumefantrine remains a highly effective partner drug for artemisinin, and no mutations or phenotypes associated with lumefantrine resistance have been observed (*Legrand et al., 2012*). High efficacy of the partner drug could therefore be an important factor limiting the spread of *pfk13* C580Y in Guyana. Finally, both the low synonymous pairwise genetic diversity ($\pi_{syn} = 3.4 \times 10^{-4}$) and the low complexity of infection (COI, 2.2% of samples with a COI > 1) in Guyana are in keeping with the expectation that the *P. falciparum* population in South America is smaller and more recently established than in Southeast Asia (*Yalcindag et al., 2012*).

The genomic structure of the Guyana parasite population could also help explain why the mutation arose there, rather than other settings in South America. The present findings indicate high outcrossing rates and few clonal lineages in Guyana (0.54% pairs with IBD >0.75). This clonal diversity is relatively high, especially within a South American context. Previous studies have found much higher proportions of non-unique haplotype backgrounds (>30% of samples) within parasite populations from Colombia (*Echeverry et al., 2013*), Peru (*Dharia et al., 2010*), and Ecuador (*Sáenz et al., 2015*; *Sáenz et al., 2017*). These prior analyses used fewer genomic markers, but the results still suggest that clonal diversity may be greater in the Guiana Shield relative to the rest of the continent, possibly driven by the high *P. falciparum* transmission level observed in mining areas (*Douine et al., 2016*; *Ministry of Public Health, 2018*). As genomic background likely plays a key role in the persistence of resistance mutations like C580Y, this haplotype richness may increase the likelihood that resistance mutations can successfully establish themselves in a permissive background. Subsequently, a low recombination rate may allow these beneficial combinations to then persist within the population. A more definitive exploration of this hypothesis, and of whether Guyana's current clonal structure represents recent demographic change or historic population subdivision, will be possible as more whole-genome sequence data become available for the continent. The slower trajectory of *pfk13* C580Y may also be due to an impaired asexual blood-stage growth rate, and the absence of compensatory mutations that have been hypothesized to aid the spread of the C580Y mutation in Southeast Asia (*Amato et al., 2018*; *Li et al., 2019*).

We chose to introduce two *pfk13* mutations by gene editing: C580Y mutation because of its presence in Guyana and its dominance in Southeast Asia, and R539T as a positive control for *in vitro*

resistance, as it exhibits the highest RSA observed in Southeast Asian strains along with I543T (*Ariey et al., 2014*; *Straimer et al., 2015*). Introducing these mutations conferred high levels of *in vitro* artemisinin resistance and both mutations negatively impacted growth rates *in vitro* in our parasite isolates from French Guiana, a neighboring country of Guyana. These growth rate differences provide a surrogate marker of fitness and do not necessarily predict *in vivo* success of the mutations, as multiple other parameters are also important including gametocyte production, impacts on transmission, and immunity. Our data show that the C580Y mutation was associated with a more severe growth defect compared to the R539T mutation in these asexual blood stage parasites. A similar finding was also recently reported in C580Y-edited isolates from the Thailand-Myanmar border (*Nair et al., 2018*), although disparate results were obtained for C580Y-edited isolates from Cambodia (*Straimer et al., 2017*). These contrasting data underline the influence of the parasite's genomic background, which potentially involves compensatory mutations that impact the overall growth. Further studies will be needed to screen for potential secondary mutations that may be offsetting these fitness costs. It will also be important to examine factors that could drive the persistence of mutant *pfk13* in Guyana despite the apparent fitness cost, including a consideration of local antimalarial drug usage and quality, transmission levels, and population movement.

The World Health Organization (WHO) has categorized countries regarding artemisinin resistance based on *pfk13* mutant prevalence and the clinical response to artemisinin derivatives (*World Health Organization, 2017*). The threshold of 5% of *pfk13* mutants has been met on several occasions in Guyana, first in 2010 (*Chenet et al., 2016*), then punctually in 2016–2017 in Region 1. The situation is evolving and a marker could disappear for reasons that are currently unclear. Nonetheless, in light of these findings, Guyana has the status of suspected resistance to artemisinin derivatives (*World Health Organization, 2017*). The clinical and public health significance of the presence of these mutations on the therapeutic efficacy of artemisinin-based combination therapies in Guyana, and more precisely artemether/lumefantrine - first line in the country, should be directly evaluated. We note that our study had limitations including the small number of analyzed samples and the *in vitro* phenotypic impact evaluated in parasites from nearby French Guiana and not Guyana itself. Nonetheless, we observed that *pfk13* mutants still circulated in 2016–2017. Given their decreased susceptibility to DHA *in vitro*, these mutants are likely to expose the partner drug to the risk of emerging resistance. This situation could defeat *P. falciparum* elimination strategies in this region. Therefore, national health authorities should continue to work together in order to overcome difficulties coming from their multiplicity of regulations, malaria control strategies and transmission levels in the context of major human migrations across borders. The Guiana Shield initiative launched by the Pan American Health Organization (PAHO) and the WHO in 2016 will have a crucial role to reinforce and to coordinate the different malaria control measures.

## Materials and methods

### Sample collection

In collaboration with PAHO, 854 *P. falciparum* isolates were collected in Guyana from March 2016 to September 2017 in febrile individuals visiting the malaria clinic. Parasites associated with these samples came mainly from Region 1 (n = 114), 7 (n = 572) and 8 (n = 150) (*Figure 1*, *Figure 1—source data 1*). Those are the three main endemic regions for malaria in the country. Sample collection was done in patients who provided informed consent. As part of molecular surveillance established in the country, consent from the patients was not required and filter papers were collected at the diagnosis time in different clinics and hospital of the country.

### DNA extraction and *pfk13* genotyping

DNA was extracted from dried blood spots using the QIAmp DNA mini kit according to the manufacturer's protocol (Qiagen, Germany). The *pfk13* gene of each sample was amplified by nested PCR using published primers (*Ariey et al., 2014*; *Menard and Ariey, 2013*). For the primary PCR, 1 µl of DNA was amplified in a Mastermix containing: 1X of HOT FIREPol Blend Master Mix Ready to Load (12.5 mM $MgCl_2$, Solis BioDyne), 3.5 mM $MgCl_2$, and 0.2 µM of each primer to a final volume of 25 µl. The amplification program was: 15 min at 95°C, then 35 cycles of 30 s at 95°C, 2 min at 60°C, 2 min at 72°C, and a final extension of 10 min at 72°C. For the nested PCR, 1 µl of primary PCR

product was amplified under the same conditions with the following program: 15 min at 95°C, then 40 cycles of 30 s at 95°C, 1 min at 60°C, 1 min at 72°C, and a final extension of 10 min at 72°C. Nested PCR products were detected using 2% agarose gel electrophoresis and ethidium bromide staining. Double-strand sequencing was performed by Eurofins (France). Sequences were aligned with Geneious v8.1.7 using the 3D7 *pfk13* sequence as a reference. Mutant isolates were analyzed twice to confirm results.

### *pfk13*-flanking microsatellite analysis

The analysis of eight microsatellite loci located upstream (−31.9,–6.36, −3.74,–0.15 kb) and down-stream (3.4, 8.6, 15.1, 72.3 kb) of the *pfk13* gene within isolates from Guyana was performed as previously described (*Chenet et al., 2016*; *Cheeseman et al., 2012*; *Talundzic et al., 2015*).

### Genetic modification of parasites at the positions 580 and 539 of the *pfk13* gene

Parasites were cultured in human red blood cells in enriched RPMI medium containing 10% human serum and were propagated at 37°C in 10% $O_2$, 5% $CO_2$ and 85% $N_2$. The enriched medium is composed of RPMI-1640 (ref 4130, Sigma Aldrich) with HEPES [25 mM], L-glutamine [5 mM], glucose [22 mM], $NaHCO_3$5% [25 mM], gentamycin [20 mM], hypoxanthine [0.37 mM] and orotic acid [1.6 µM]. Two *P. falciparum* isolates collected in French Guiana in 2011 (O141-A) and 2014 (R086) were genetically modified using the ZFN method as previously described (*Straimer et al., 2015*). Briefly, donor plasmids (pZFN-K13-18/20-hDHFR-bsmut carrying the wild-type *pfk13* allele and pZFN-K13-18/20-hDHFR-**C580Y** and pZFN-K13-18/20-hDHFR-**R539T** carrying the mutated *pfk13* allele) were purified from XL10-Gold bacteria using the QIAGEN plasmid Maxi kit (ref12162) and resuspended in Cytomix. Parasites were electroporated with 50 µg of donor plasmid using the Biorad Gene-PulserII electroporator with settings of 0.31 kV and 950 µF (*Fidock and Wellems, 1997*). The day after electroporation and for 6 days, parasites were exposed to 2.5 nM WR99210 (a gift from Jacobus Pharmaceuticals, Princeton, NJ). Between 20–35 days after electroporation, parasites were detectable by microscopy. To check plasmid integration into parasites, DNA was extracted from bulk culture with the QIAamp DNA Mini Kit (Qiagen) and was PCR amplified using primers p16 (5'-GCTAA TAAGTAATATCAATATAAGGG-3') and p17 (5'-GGTATTAAATTTTTACCATTCCCATTAGTATTTTG TATAGG-3'). Sequencing was done by Eurofins with the p16 primer and sequences were analyzed using Geneious v8.1.7. Edited parasites were cloned by limiting dilution and selected after 3 weeks by lactate dehydrogenase assay using Malstat reagent (*Goodyer and Taraschi, 1997*; *Adjalley et al., 2010*). Selected clones were *pfk13* genotyped.

### *In vitro* drug sensitivity assays (ring-stage survival assay)

Parental and *pfk13*-edited parasites were phenotyped using the Ring-stage Survival Assay ($RSA_{0-3h}$) method as previously described (*Witkowski et al., 2013*). $RSA_{0-3h}$ were performed independently three times for each line. The *pfk13* I543T-mutant IPC4912 strain (MRA1241) was used as an artemisinin-resistant control. $RSA_{0-3h}$ was interpretable if the initial parasitemia was greater than 0.25% and if the growth rate was greater than two-fold per 48 hr. Statistical significance between survival rates of the different lines were calculated using Student's t-test.

### Competitive growth fitness assays

Fitness assays were performed by inoculating an equal number of wild-type and mutant isogenic ring-stage parasites in a 6 ml culture at an initial parasitemia of 1%. Each co-culture experiment was performed in duplicate on two independent occasions and monitored for 60 days. Saponin-lysed parasite pellets of each co-culture were harvested every two days during 60 days and genomic DNA was extracted using the QIAmp DNA mini kit. The percentage of wild-type or mutant allele in each co-culture was determined using genomic DNA in a TaqMan Allelic Discrimination Real-Time PCR Assays. Wilcoxon signed-rank test was applied to test for statistical significance in the observed percentage of the *pfk13* mutant allele in each of the individual paired co-cultures against an expected no change in the percentage of the mutant.

## TaqMan allelic discrimination real-time PCR (qPCR) assays

Primers (forward and reverse) and TaqMan fluorescence-labeled MGB probes (Eurofins, Germany) for real-time quantitative PCR (qPCR) were designed to specifically detect the *pfk13* propeller mutations R539T and C580Y, or the ZFN silent binding mutations, in parasites. The sequences of the forward and reverse primers, and probes (FAM and HEX probe) are shown in *Supplementary file 4*. We first determined the efficiency and sensitivity of amplifying the *pfk13* allele in real-time PCR assays using standard curves comprising 10-fold serially diluted DNA templates ranging from 10 ng to 0.001 ng. Robustness was demonstrated by high efficiency (88–95%) and $R^2$ values (0.98–1.00) (*Figure 5—figure supplement 1a and b*, *Figure 5—figure supplement 1—source data 1*). Next, we tested the quantitative accuracy in genotype calling by performing multiplex qPCR assays using a pre-defined set of mixtures containing plasmids expressing wild-type to mutant alleles in fixed ratios (0:100, 20:80, 40:60, 50:50, 60:40, 80:20, 100:0). The triplicate points clustered tightly, indicating high reproducibility in the data across the fitted curve (linear polynomial order = 2, $R^2$ = 0.92 to 0.94) (*Figure 5—figure supplement 1c and d*, *Figure 5—figure supplement 1—source data 1*). Hence, our assay was able to accurately quantify the different proportions of the mutant alleles in these pre-mixed samples. For a single set of samples, we ran the qPCR on two separate occasions to test the consistency of amplification between runs. The results showed excellent correlations (*Figure 5—figure supplement 2a*, *Figure 5—figure supplement 2—source data 1*), and thus we concluded that it was not necessary to perform qPCR in replicate runs for each sample. Nevertheless, we always included three replicate qPCR reactions per sample in every run. The data showed consistently high correlations between independent sampling experiments (*Figure 5—figure supplement 2b*, *Figure 5—figure supplement 2—source data 1*).

Purified DNA templates were amplified with a species-specific primer set and the corresponding probe. Briefly, the qPCR reactions for every sample were run in triplicates consisting of 1x QuantiFAST reaction mix containing ROX reference dye (Qiagen, Germany), 0.66 µM of forward and reverse primers, 0.16 µM each of the FAM-MGB and HEX-MGB TaqMan probes, and 10 ng of genomic DNA. Amplification and detection of fluorescence was carried out on the QuantStudio 3 (Applied Biosystems, USA) using the genotyping assay mode with the following cycling conditions: 30 s at 60℃, 5 min at 95℃ to activate the enzyme, and 40 cycles of 30 s at 95℃, 30 s at 60℃, and 30 s at 60℃ and post-read out at 60℃ for 30 s. The final optimized assays always included a positive control of plasmid expressing the mutant or wild-type allele in every PCR reaction to ensure the reaction was successful and a no template control of water as a negative control.

The software analyzed the background fluorescence level and calculated background normalized media dye fluorescence (ΔRn) as a function of cycle number for the wild-type or mutant allele. To determine the wild-type or mutant allele frequency in each sample, we first confirmed the presence of the allele by only taking values where the $C_t$ of sample was less than that of the no template control minus three cycles. Next, we subtracted the sample's ΔRn from the background (control plasmid: absence of wild-type or mutant expressing allele) and normalized to 100% (control plasmid: 100% wild-type or mutant expressing allele) to obtain the percentage of mutant and wild-type allele. Then we calculated the average of (mutant and 100% - wild-type) to derive the percentage of mutant allele.

## Whole-genome sequencing and variant calling

For the Guyana samples, we performed selective whole-genome amplification (SWGA) on DNA samples as previously described (*Oyola et al., 2016*) to enrich parasite DNA prior to sequencing on an Illumina HiSeqX instrument at the Broad Institute. We used the enriched DNA to construct Illumina sequencing libraries from the amplified material using the Nextera XT library kit (catalog no. FC-131–1002). We aligned reads to the *P. falciparum* v3 reference genome assembly using BWA-MEM (*Li, 2013*) and called SNPs and INDELs using the GATK HaplotypeCaller (*Van der Auwera et al., 2013*; *McKenna et al., 2010*; *DePristo et al., 2011*) according to the best practices for *P. falciparum* as determined by the Pf3K consortium (https://www.malariagen.net/projects/pf3k). Analyses were limited to the callable segments of the genome (*Miles et al., 2016*) and excluded sites where over 20% of samples were heterozygous. Additional BAM files for comparative analyses among populations were downloaded from the Pf3k project (release 5; www.malariagen.net/projects/pf3k). For each of four countries in Africa (Democratic Republic of Congo, Ghana, Guinea, and Malawi) and

two countries in Southeast Asia (Cambodia and Thailand), 50 samples were chosen based on their high coverage (greatest number of sites with at least $10 \times$ coverage).

## Genomic analysis

We calculated pairwise identity by descent (IBD) using a hidden Markov model (hmmIBD) (*Schaffner et al., 2018*). Samples with a missing call rate >0.7 were excluded from the analysis. We conducted PCA analyses using R (*R Development Core Team, 2016*) after removing samples that showed high relatedness (IBD >0.5) to another sample with higher sequencing coverage. Pairwise nucleotide diversity was calculated using a custom Perl script (Source Code File 1, Source Code File 2, Source Code File 3). All genomic analyses were limited to samples containing only a single clonal lineage, as determined by THE REAL McCOIL (*Chang et al., 2017*) or as identified by the Pf3K consortium (*Zhu et al., 2019*).

## Acknowledgements

LM gratefully acknowledges funding support from Global Malaria Program (World Health Organization), French Ministry for research, European Commission Grant (Regional fund for Development, Synergie GY0012082), Santé Publique France as National Reference Center for Malaria and Investissement d'Avenir grant managed by Agence Nationale de la Recherche (CEBA, ref ANR-10-LABX-25–01). DAF gratefully acknowledges funding support from the NIH (R01 AI109023; R01 124678 and R37 AI50234) and the Bill and Melinda Gates Foundation (OPP1201387). DEN and AME have been funded in part with Federal funds from the National Institute of Allergy and Infectious Diseases, National Institutes of Health, Department of Health and Human Services, under Grant Number U19AI110818 to the Broad Institute. SM receives support from a Human Frontiers Science Program Long-Term Fellowship. We thank Leila Ross and Barbara Stokes for their help with *in vitro* fitness assays.

## Additional information

### Competing interests

Maria-Paz Ade, Jean SF Alexandre, Pascal Ringwald: MPA, JSFA, and PR are staff members of the World Health Organization. The authors alone are responsible for the views expressed in this publication and they do not necessarily represent the decisions, policy or views of the World Health Organization. The other authors declare that no competing interests exist.

### Funding

| Funder | Grant reference number | Author |
| --- | --- | --- |
| European Commission | Synergie GY0012082 | Luana C Mathieu |
| Sante Publique France | NRC for malaria | Yassamine Lazrek<br>Lise Musset |
| Agence Nationale de la Recherche | ANR-10-LABX-25-01 | Lise Musset |
| Human Frontier Science Program | Long-Term Fellowship | Sachel Mok |
| Bill and Melinda Gates Foundation | OPP1201387 | David Fidock |
| National Institutes of Health | R01 AI109023 | David Fidock |
| National Institute of Allergy and Infectious Diseases | U19AI110818 | Angela M Early<br>Daniel E Neafsey |
| World Health Organization | Global Malaria Program | Lise Musset |
| National Institutes of Health | R01 124678 | David Fidock |
| National Institutes of Health | R37 AI50234 | David Fidock |

| U.S. Department of Defense | W81XWH1910086 | David Fidock |

The funders had no role in study design, data collection and interpretation, or the decision to submit the work for publication.

## Author contributions

Luana C Mathieu, Sachel Mok, Conceptualization, Data curation, Formal analysis, Validation, Investigation, Methodology, Writing - original draft, Writing - review and editing; Horace Cox, Conceptualization, Resources, Methodology, Project administration, Writing - review and editing, Coordinate sample collection; Angela M Early, Conceptualization, Software, Formal analysis, Validation, Investigation, Methodology, Writing - original draft, Writing - review and editing; Yassamine Lazrek, Supervision, Investigation, Project administration; Jeanne-Celeste Paquet, Data curation, Investigation; Maria-Paz Ade, Conceptualization, Supervision, Funding acquisition, Methodology, Project administration, Writing - review and editing, Coordinate sample collection; Naomi W Lucchi, Data curation, Investigation, Methodology, Writing - review and editing; Quacy Grant, Conceptualization, Supervision, Project administration; Venkatachalam Udhayakumar, Formal analysis, Supervision, Validation, Investigation, Methodology; Jean SF Alexandre, Resources, Supervision, Funding acquisition, Project administration, Coordinate sample collection; Magalie Demar, Supervision, Project administration; Pascal Ringwald, Conceptualization, Resources, Formal analysis, Supervision, Validation, Project administration, Writing - review and editing; Daniel E Neafsey, Conceptualization, Resources, Data curation, Formal analysis, Funding acquisition, Investigation, Methodology, Writing - review and editing; David A Fidock, Conceptualization, Resources, Formal analysis, Supervision, Funding acquisition, Validation, Investigation, Methodology, Writing - review and editing; Lise Musset, Conceptualization, Resources, Formal analysis, Supervision, Funding acquisition, Validation, Investigation, Methodology, Writing - original draft, Project administration, Writing - review and editing

## Author ORCIDs

Luana C Mathieu (iD) https://orcid.org/0000-0002-6021-468X
Sachel Mok (iD) https://orcid.org/0000-0002-9605-0154
Jeanne-Celeste Paquet (iD) https://orcid.org/0000-0001-6046-0154
Daniel E Neafsey (iD) https://orcid.org/0000-0002-1665-9323
David A Fidock (iD) https://orcid.org/0000-0001-6753-8938
Lise Musset (iD) https://orcid.org/0000-0003-0215-4110

## Ethics

Human subjects: The samples were collected as part of the routine surveillance system implemented in Guyana. Health care facilities were in charge of collecting anonymized *P. falciparum* positive cases. Identification of individuals cannot be established. In accordance with WHO guidelines on ethical issues in public health surveillance the sample collection in Guyana was exempt of ERC since this intervention was part of the Malaria control program defined by the Ministry of Public Health of the country as monitoring of public health programs (https://www.who.int/ethics/publications/public-health-surveillance/en/). The analysis of the samples was also approved by the Environmental Protection Agency in the frame on the Nagoya Protocol on Access to Genetic Resources and the Fair and Equitable Sharing of Benefits Arising from their Utilization.

## Decision letter and Author response

Decision letter https://doi.org/10.7554/eLife.51015.sa1
Author response https://doi.org/10.7554/eLife.51015.sa2

# Additional files

## Supplementary files

• Source code 1. Source code to calculate the nucleotide diversity per nucleotide.

- Source code 2. Source code to calculate the nucleotide diversity per genomic region.

- Source code 3. Example of input file to do calculations.

- Supplementary file 1. Resistance genotype profiles of parasites from Guyana and French Guiana.

- Supplementary file 2. Guyana SWGA associated markers.

- Supplementary file 3. Comparison of synonymous nucleotide diversity per country.

- Supplementary file 4. List of the sets of forward and reverse primers and dual fluorescent-labeled FAM/HEX MGB probes used.

- Transparent reporting form

## Data availability

The authors declare that the data supporting the findings of this study are available within the paper and its supplementary information. Whole genome sequencing data from this study are available from the NCBI Sequence Read Archive under BioProject ID PRJNA543530. Genome data from the Pf3K project are available at https://www.malariagen.net/projects/pf3k.

The following dataset was generated:

| Author(s) | Year | Dataset title | Dataset URL | Database and Identifier |
|---|---|---|---|---|
| Early A, Mathieu L, Cox H, Musset L, Neafsey DE | 2018 | Whole Genome Sequencing of P. falciparum collecte in Guyana, 2016-2017 | https://www.ncbi.nlm.nih.gov/bioproject/?term=PRJNA543530 | NCBI BioProject, PRJNA543530 |

The following previously published dataset was used:

| Author(s) | Year | Dataset title | Dataset URL | Database and Identifier |
|---|---|---|---|---|
| Kwiatkowski DP | 2019 | MALARIAGEN: Malaria Genomic Epidemiology Network | https://www.malariagen.net/projects/pf3k | Malaria Gen, Pf3k version 6 |

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
