## [Decision Letter]

**Acceptance summary:**

This molecular epidemiological study describes a set of *Plasmodium falciparum* samples collected in Guyana that contain the K13 propeller domain mutation most commonly associated with artemisinin resistance in Southeast Asia. The authors show conclusively that mutations associated with artemisinin resistance identified in Guyana are indeed local and belong to a single lineage. They then proceed with an in vitro characterization of parasites from a proximal geographic location (French Guiana) and use those findings to infer the clinical profile of the mutant parasites. This constitutes an important study documenting potential evolution of artemisinin resistance in another continent under the extensive use of artemisinin combined therapy.

**Decision letter after peer review:**

Thank you for submitting your article "A local emergence and unusual evolution of artemisinin resistance of *Plasmodium falciparum* in eastern Amazonia" for consideration by *eLife*. Your article has been reviewed by three peer reviewers, and the evaluation has been overseen by Neil Ferguson as the Senior Editor. The following individuals involved in review of your submission have agreed to reveal their identity: Liwang Cui (Reviewer #3).

The reviewers have discussed the reviews with one another and the Reviewing Editor has drafted this decision to help you prepare a revised submission.

Summary:

This manuscript describes a set of *P. falciparum* samples collected in Guyana that contain the K13 propeller domain mutation most commonly associated with artemisinin resistance in SE Asia. Genetic evaluation showed that Guyanese parasites with the K13 mutation were closely related to each other, but not to SE Asian resistant clones. The mutation was shown to mediate resistance (delayed clearance in culture) when introduced into parasites with a related genetic background (from French Guiana) and to be associated with slightly decreased fitness in an in vitro assay. While we think that this is a well-written, and valuable report, we have some concerns about the way the results are interpreted and the conclusions derived from them.

Essential revisions:

Two key limitations of the paper are the lack of parasites from Guyana and the lack of clinical data. The authors make good efforts to work around these limitations, but nevertheless they affect the relevance of the findings. Authors should considerably reassess the impact of their conclusions. In particular:

1) In vitro fitness costs do not necessarily correlate with in vivo success of the mutations. As stated by the authors, in vitro fitness analysis of *kelch13* mutations have historically shown a strong dependency on the background even within the same country (Cambodia) and, given the caveat above, I'm not sure how to interpret them here. However, competition assays seem to consistently show a higher fitness cost for C580Y mutations compared to others. Interestingly, that's the exact opposite of that has been observed in the field, with parasites carrying the C580Y variant rapidly spreading and the R539T slowly disappearing. This is also somehow implied in this work by the fact that C580Y mutations are still circulating after many years.

According to WHO's guidelines (e.g. https://apps.who.int/iris/bitstream/handle/10665/274362/WHO-CDS-GMP-2018.18-eng.pdf?sequence=1&isAllowed=y), C580Y is one of the markers recognised for molecular surveillance. However, by the same guidelines, TESs are still the recommended tool for to confirm artemisinin resistance, which according to the authors, have found no evidence of delayed clearance in the country so far. The authors should reassess the impact of the conclusions and word the paper more cautiously (even the title) avoiding any inference about the clinical impact of these data (Discussion paragraph five).

2) It has already been shown that C580Y causes increases in RSA values. We understand the idea of testing this hypothesis on another background (with the caveats above), but, again, it has been amply demonstrated that there is no “in vitro” effect of the background. In fact, even 3D7+C80Y, despite having no *crt* mutations whatsoever, has been shown to have elevated RSA values. It would be good to clarify what the key message of these experiments is. Also, it is unclear why the authors also decided to test R539T, given that the variant appears to be disappearing. Why not Y493H, which on the other end is becoming more frequent?

3) The authors decide to edit parasites from a different geographical location. While we understand that this might be due to unstated practical constraints (e.g. the original isolates from Guyana not being available), I believe any conclusion should be taken with a pinch of salt and without any inferential jump. For example, the authors use PCA to infer that parasites from Guyana and French Guiana are genetically similar, but I'm not sure how they quantified that statement. PCA is influenced by many factors, including, but not limited to, genetic similarity. I think it's important to see genetic similarity measured directly and perhaps use publicly available samples to put that measure into context: are those parasites as similar as two parasites from Guyana, from Asia, or from Africa?

4) Related to the points above, it would be helpful for the authors to include a paragraph addressing limitations in their description. First, clinical delayed clearance after treatment with artemisinins has not been described in Guyana. Second, the number of mutant parasites studied was fairly small, and whole genome sequencing led to some incomplete results. Third, gene editing to link genotypes with phenotypes did not utilize parasites from Guyana, although they were appropriately from a nearby region.

[Editors' note: further revisions were suggested prior to acceptance, as described below.]

Thank you for resubmitting your work entitled "Local emergence in Amazonia of *Plasmodium falciparum k13* C580Y mutants associated with in vitro artemisinin resistance" for further consideration by *eLife*. Your revised article has been evaluated by Neil Ferguson as SeniorEditor.

Please could you modify the Discussion a little more to take into account the concerns of reviewer 2 below, at least to some extent, then resubmit. Further review will not then be needed.

Reviewer #1:

The authors have responded appropriately to review requests.

Reviewer #2:

I'd like to thank the authors for the all the changes and additional analysis. As a result I believe they've really added solidity to the work and made the presentation more compelling.

I feel satisfied that all points raised were addressed to satisfaction, apart from the request to reword their conclusions more cautiously. I still struggle, in fact, with the over-interpretation of the results and their public health implications. I do believe that these results should be urgently shared with the community but I also believe that WHO's guidelines are in place exactly to avoid misinterpretations and to provide an objective depiction of the situation.

More specifically:

Discussion: "In light of these findings, according to WHO definitions, Guyana has the status of suspected resistance to artemisinin derivatives."

I think this statement is at very least misleading, perhaps incorrect. WHO guidelines, in the official policy cited, state that "Suspected endemic artemisinin resistance is defined as: (among other criteria) ≥ 5% of patients carrying K13 resistance-confirmed mutations (listed in Table 1 (NB: which includes C580Y))". The authors found in their data a C580Y prevalence of 1.6% (which they themselves define as "low") which (luckily!) doesn't meet the criteria. It is true that in Region 1 (and in Region 1 only) the prevalence goes over 5% and reaches 8.8% but that result can't be generalised to the whole country. (On a side note, it's interesting that Region 1 is on the border with Venezuela, where there has been a recent outbreak of cases). I accept that there may be data specific to support that statement but that should be clearly stated and can't be attributed to the findings in this study.

Discussion: "The clinical impact of the *pfk13* C580Y mutation on the therapeutic efficacy of artemisinin-based combination therapies and more precisely artemether/lumefantrine, the first line in Guyana, should be evaluated to test for delayed clearance."

It might be semantic but I think there is an important subtlety here. I don't think anyone would argue with the amply demonstrated clinical impact of *pfk13* C580Y mutations per se. The question is on the impact of the mutations observed in Guyana on the clinical phenotype. I think the authors should be rephrase their statement more directly, e.g. along the lines of "The clinical and public health significance of the presence of these mutations on the therapeutic efficacy of artemisinin-based combination therapies in Guyana, and more precisely artemether/lumefantrine – first line in the country, should be directly evaluated."

The reality is that the data presented are very interesting and sufficient to at very least raise an alarm but not to derive any definitive conclusion about the spread of artemisinin resistance. It goes without saying that the situation needs to be monitored very closely and this work offers a fantastic baseline for the future.

Reviewer #3:

All previous concerns and comments have been properly addressed.

---

## [Author Response]

Essential revisions:Two key limitations of the paper are the lack of parasites from Guyana and the lack of clinical data. The authors make good efforts to work around these limitations, but nevertheless they affect the relevance of the findings. Authors should considerably reassess the impact of their conclusions. In particular:1) In vitro fitness costs do not necessarily correlate with in vivo success of the mutations. As stated by the authors, in vitro fitness analysis of kelch13 mutations have historically shown a strong dependency on the background even within the same country (Cambodia) and, given the caveat above, I'm not sure how to interpret them here. However, competition assays seem to consistently show a higher fitness cost for C580Y mutations compared to others. Interestingly, that's the exact opposite of that has been observed in the field, with parasites carrying the C580Y variant rapidly spreading and the R539T slowly disappearing. This is also somehow implied in this work by the fact that C580Y mutations are still circulating after many years.According to WHO's guidelines (e.g. https://apps.who.int/iris/bitstream/handle/10665/274362/WHO-CDS-GMP-2018.18-eng.pdf?sequence=1&isAllowed=y), C580Y is one of the markers recognised for molecular surveillance. However, by the same guidelines, TESs are still the recommended tool for to confirm artemisinin resistance, which according to the authors, have found no evidence of delayed clearance in the country so far. The authors should reassess the impact of the conclusions and word the paper more cautiously (even the title) avoiding any inference about the clinical impact of these data (Discussion paragraph five).

Thank you for these suggestions. We have carefully reworded the manuscript to clarify these points. The Title has been changed to “Local emergence in Amazonia of *Plasmodium falciparum k13* C580Y mutants associated with in vitro artemisinin resistance.”

In the Introduction, we have introduced the concept of partial resistance to artemisinins as defined by the WHO.

Introduction: “Resistance to artemisinin is partial and affects only rings (World Health Organization, 2018). Clinically, this partial resistance trait manifests as a parasite clearance half-life that exceeds 5.5 hours (WWARN K13 Genotype-Phenotype Study Group, 2019). This half-life represents the time required to achieve a two-fold reduction of the parasite biomass. Partial resistance also manifests as persistent parasitemia on day three with a complete clearance of parasites following full treatment with an artesunate monotherapy lasting seven days or with an ACT (World Health Organization, 2018).”

The Discussion also emphasizes that the clinical impact should be assessed, as we do not know the impact of this mutation on the clinical efficacy of artesunate or ACTs: “Further studies will be needed to screen for potential secondary mutations that may be offsetting these fitness costs. It will also be important to examine factors that could drive the persistence of mutant *pfk13* in Guyana despite the apparent fitness cost, including a consideration of local antimalarial drug usage and quality, transmission levels, and population movement.”

Discussion: “In light of these findings,according to WHO definitions, Guyana has the status of suspected resistance to artemisinin derivatives (World Health Organization, 2017). The clinical impact of the *pfk13* C580Y mutation on the therapeutic efficacy of artemisinin-based combination therapies and more precisely artemether/lumefantrine, the first line in Guyana, should be evaluated to test for delayed clearance.”

“In vitro fitness costs do not necessarily correlate with in vivo success of the mutations.”

We absolutely agree. Our study evaluated the capacity of the mutant to compete with wild-type parasites in culture. This is an artificial evaluation of the fitness cost, in the absence of drug or other parameters important in the in vivo context (including gametocyte production, impacts on transmission, immunity). We recognize that our in vitro findings are therefore only one component of the overall fitness of the parasite.

Using this parameter, our study and previous data reported an impact of the C580Y mutation on parasite growth in vitro, when competing with wild-type isogenic parasites, that clearly depends on the genetic background. In our study, C580Y had a detectable fitness cost in both the O141-A and R086 strains. By comparison, earlier fitness studies showed that the C580Y mutation had a negligible impact on fitness in strains recently culture adapted from Cambodia, contrasting with a substantial fitness cost in the V1/S strain that was adapted to culture decades ago (Straimer et al., 2017). In a separate study, the C580Y mutation had a fitness cost when introduced into an artemisinin-sensitive Thai strain (Nair et al., 2018).

These points are addressed in the manuscript Discussion section, as follows: “These growth rate differences provide a surrogate marker of fitness and do not necessarily predict in vivo success of the mutations, as multiple other parameters are also important including gametocyte production, impacts on transmission, and immunity. Our data show that the C580Y mutation was associated with a more severe growth defect compared to the R539T mutation in these asexual blood stage parasites. A similar finding was also recently reported in C580Y-edited isolates from the Thailand-Myanmar border (Nair et al., 2018), although disparate results were obtained for C580Y-edited isolates from Cambodia (Straimer et al., 2017). These contrasting data underline the influence of the parasite’s genomic background, which potentially involves compensatory mutations that impact the overall growth.”

In SEA, C580Y parasites are spreading despite the fitness cost observed in some strains, presumably in part because of substantial selection pressure applied to the infected patient population. In many areas of the Greater Mekong Sub-region in SEA, *pfk13* C580Y mutants also often carry additional mutations (outside of the *pfk13* locus) that allow them to resist to the partner drug, piperaquine. This is a very important driver to expand the population of mutant parasites despite the *K13* fitness impact. The situation in Guyana is different as artemether/lumefantrine is the major ACT and no resistance to lumefantrine has been described.

Please refer to the Discussion section of the manuscript, as follows: “In the Guiana Shield, lumefantrine remains a highly effective partner drug for artemisinin, and no mutations or phenotypes associated with lumefantrine resistance have been observed (Legrand et al., 2012). High efficacy of the partner drug could therefore be an important factor limiting the spread of *pfk13* C580Y in Guyana.”

To better understand why *pfk13* mutant parasites are still circulating in the region after several years in spite of the apparent fitness cost of the mutation, more investigation is needed to identify the factors that drive persistence. These factors could include antimalarial drug usage, genetic background specific to this country, population movement, and the level of transmission.

Our revised Discussion addresses this as follows: “However, the situation in this part of the world is slightly different from Southeast Asia. First, efforts to monitor *pfk13* mutations in Guyana were not systematically conducted between 2010 and 2017. Sample size could therefore explain why the C580Y mutation was only sporadically observed. Nonetheless, the present study indicates that the mutation has not drastically increased in the parasite population.”

“Further studies will be needed to screen for potential secondary mutations that may be offsetting these fitness costs. It will also be important to examine factors that could drive the persistence of mutant *pfk13* in Guyana despite the apparent fitness cost, including a consideration of local antimalarial drug usage and quality, transmission levels, and population movement.”

2) It has already been shown that C580Y causes increases in RSA values. We understand the idea of testing this hypothesis on another background (with the caveats above), but, again, it has been amply demonstrated that there is no “in vitro” effect of the background.

The effect of the parasite genetic background on in vitro sensitivity to artemisinins was demonstrated by Straimer et al., 2015, who showed that the C580Y mutation conferred a RSA survival rate that ranged from 1.9% to 24.1% when introduced into five separate strains. This effect was even more important when the strains were genetically very different (e.g. FCB versus Cam3.II). To limit this caveat, we have chosen to genetically modify the only available strains from the same region, namely two strains adapted from French Guiana. To strengthen these results, as suggested by the reviewers, we have improved the demonstration of similarities between French Guianan and Guyanese strains (please refer to our reply to point 3 below).

In fact, even 3D7+C80Y, despite having no crt mutations whatsoever, has been shown to have elevated RSA values. It would be good to clarify what the key message of these experiments is. Also, it is unclear why the authors also decided to test R539T, given that the variant appears to be disappearing. Why not Y493H, which on the other end is becoming more frequent?

The key message of this experiment was to evaluate the impact of *pfk13* mutations on the in vitro phenotype to artemisinin. We chose to introduce two mutations by gene editing: C580Y because of its presence in Guyana and its prevalence in Southeast Asia and R539T as a positive control of in vitro resistance (as it produced a higher RSA survival phenotype than C580Y across multiple Southeast Asian strains (Ariey et al., 2014, Straimer et al., 2015). Our 2016 report on *pfk13* mutations in SEA (Menard et al., 2016) showed a similar prevalence of R539T and Y493H, with the latter conferring less resistance in gene-edited parasites (e.g. in *pfk13* edited Dd2 parasite clones we observed RSA values of 1.7% for Y493H and 19.4% for R539T; Straimer et al., 2015). We have also examined the *pfk13* data in the WWARN molecular surveyor database. Of 1992 genomes analyzed we observed 62 with R539T and 39 with Y493H. Of the 394 genomes reported for the period 2015-2017 (primarily from the first two years), we note that 6 had R539T (from 2015 and 2016) and 3 had Y493H (all from 2015). Thus, we do not see clear evidence in the literature or the WWARN database analysis that Y493H is expanding. There may be data specific to a certain region that has not been published and therefore could not inform our selection of mutations. Our results also demonstrated that the in vitro impact of these two mutations was similar to the one observed on Cambodian strains, i.e. we observed a higher cost of the R539T mutation along with high survival rates.

These details have been added to the Discussion: “The slower trajectory of *pfk13* C580Y may also be due to an impaired asexual blood-stage growth rate, and the absence of compensatory mutations, which have been hypothesized to aid the spread of the C580Y mutation in Southeast Asia (Amato et al., 2018, Li et al., 2019). We chose to introduce two *pfk13* mutations by gene editing: C580Y mutation because of its presence in Guyana and its dominance in Southeast Asia, and R539T as a positive control for in vitroresistance, as it exhibits the highest RSA observed in Southeast Asian strains along with I543T (Ariey et al., 2013, Straimer et al., 2015).”

3) The authors decide to edit parasites from a different geographical location. While we understand that this might be due to unstated practical constraints (e.g. the original isolates from Guyana not being available), I believe any conclusion should be taken with a pinch of salt and without any inferential jump. For example, the authors use PCA to infer that parasites from Guyana and French Guiana are genetically similar, but I'm not sure how they quantified that statement. PCA is influenced by many factors, including, but not limited to, genetic similarity. I think it's important to see genetic similarity measured directly and perhaps use publicly available samples to put that measure into context: are those parasites as similar as two parasites from Guyana, from Asia, or from Africa?

This is a very valid point, thank you for bringing it to our attention. We have included a new supplemental figure (Figure 2—figure supplement 1) that depicts the absolute rate of SNP differences between isolates from the samples utilized in the PCA. This new figure shows that while pairwise comparisons of parasites reveals moderately higher divergence between French Guiana and Guyana than within Guyana (indicating slight genetic divergence between the populations), there is greater genetic similarity between parasites from Guyana and French Guiana than is typically observed for pairs of parasites from the same country in other parts of the world, notably Cambodia, Thailand, Ghana, Malawi, the Democratic Republic of Congo, and Guinea. We thank the reviewer for helping us to clarify this important point.

4) Related to the points above, it would be helpful for the authors to include a paragraph addressing limitations in their description. First, clinical delayed clearance after treatment with artemisinins has not been described in Guyana. Second, the number of mutant parasites studied was fairly small, and whole genome sequencing led to some incomplete results. Third, gene editing to link genotypes with phenotypes did not utilize parasites from Guyana, although they were appropriately from a nearby region.

Our revised Discussion now states, the two major limitations of our study (namely the small number of genotyped parasites and the construction of isogenic lines in French Guianan but not Guyanese strains). Other limitations have also been stated in other paragraphs of the Discussion. Regarding the absence of delayed parasite clearance times in Guyana, we note that a therapeutic efficacy study has been conducted only once in this region. However, the sample size was low and did not include a C580Y carrier. The impact of the *pfk13* C580Y mutation on therapeutic response to confirm the status of the country still needs to be assessed. This task will be difficult as the mutation is rare and therefore the probability to observe a C580Y infection among included patients is very low.

Discussion: “The clinical impact of the *pfk13* C580Y mutation on the therapeutic efficacy of artemisininbased combination therapies and more precisely artemether/lumefantrine, the first line in Guyana, should be evaluated to test for delayed clearance. We note that our study had limitations including the small number of analyzed samples and the in vitrophenotypic impact evaluated in parasites from nearby French Guiana and not Guyana itself. Nonetheless, we observed that *pfk13* mutants still circulated in 2016-2017. Given their decreased susceptibility to DHA in vitro, these mutants are likely to expose the partner drug to the risk of emerging resistance.”

[Editors' note: further revisions were suggested prior to acceptance, as described below.]

Please could you modify the Discussion a little more to take into account the concerns of reviewer 2 below, at least to some extent, then resubmit. Further review will not then be needed.Reviewer #2:I'd like to thank the authors for the all the changes and additional analysis. As a result I believe they've really added solidity to the work and made the presentation more compelling.I feel satisfied that all points raised were addressed to satisfaction, apart from the request to reword their conclusions more cautiously. I still struggle, in fact, with the over-interpretation of the results and their public health implications. I do believe that these results should be urgently shared with the community but I also believe that WHO's guidelines are in place exactly to avoid misinterpretations and to provide an objective depiction of the situation.More specifically:Discussion: "In light of these findings, according to WHO definitions, Guyana has the status of suspected resistance to artemisinin derivatives."I think this statement is at very least misleading, perhaps incorrect. WHO guidelines, in the official policy cited, state that "Suspected endemic artemisinin resistance is defined as: (among other criteria) ≥ 5% of patients carrying K13 resistance-confirmed mutations (listed in Table 1 (NB: which includes C580Y))". The authors found in their data a C580Y prevalence of 1.6% (which they themselves define as "low") which (luckily!) doesn't meet the criteria. It is true that in Region 1 (and in Region 1 only) the prevalence goes over 5% and reaches 8.8% but that result can't be generalised to the whole country. (On a side note, it's interesting that Region 1 is on the border with Venezuela, where there has been a recent outbreak of cases). I accept that there may be data specific to support that statement but that should be clearly stated and can't be attributed to the findings in this study.

The threshold of 5% of *pfk13* mutants defined by WHO to classify countries regarding artemisinin resistance has been met on several occasions in Guyana, first in 2010, then in 2016-2017 where in Region 1 the threshold has been punctually overreached. This classification is at the country scale even if in 2016-2017 only Region 1 met the threshold. In 2010, the proportion was reached on a sample set of 98 samples collected in different region of the country. Mutants were observed in Region 1 and 7, mainly in Region 7. The situation is evolving and a marker could disappear or even increase for reasons that are currently unclear but in light of these findings and criteria, six years apart, we could not avoid to mention that Guyana has the status of suspected resistance to artemisinin derivatives.

Discussion: “WHO has categorized countries regarding artemisinin resistance based on *pfk13* mutant prevalence and the clinical response to artemisinin derivatives (World Health Organization, 2017). The threshold of 5% of *pfk13* mutants has been met on several occasions in Guyana, first in 2010 (Chenet et al., 2016), then punctually in 2016-2017 in Region 1. The situation is evolving and a marker could disappear for reasons that are currently unclear. Nonetheless, in light of these findings, Guyana has the status of suspected resistance to artemisinin derivatives (World Health Organization, 2017).”

Discussion: "The clinical impact of the pfk13 C580Y mutation on the therapeutic efficacy of artemisinin-based combination therapies and more precisely artemether/lumefantrine, the first line in Guyana, should be evaluated to test for delayed clearance."It might be semantic but I think there is an important subtlety here. I don't think anyone would argue with the amply demonstrated clinical impact of pfk13 C580Y mutations per se. The question is on the impact of the mutations observed in Guyana on the clinical phenotype. I think the authors should be rephrase their statement more directly, e.g. along the lines of "The clinical and public health significance of the presence of these mutations on the therapeutic efficacy of artemisinin-based combination therapies in Guyana, and more precisely artemether/lumefantrine – first line in the country, should be directly evaluated."

We thank the reviewer for this suggestion. The sentence has been added to replace the previous one as follows: “The clinical and public health significance of the presence of these mutations on the therapeutic efficacy of artemisinin-based combination therapies in Guyana, and more precisely artemether/lumefantrine – first line in the country, should be directly evaluated.”